# SyncFlow: Temporally Aligned Joint Audio-Video Generation from Text

## Abstract

Video and audio are closely correlated modalities that humans naturally perceive together. While recent advancements have enabled the generation of audio or video from text, producing both modalities simultaneously still typically relies on either a cascaded process or multi-modal contrastive encoders. These approaches, however, often lead to suboptimal results due to inherent information losses during inference and conditioning. In this paper, we introduce SyncFlow, a system that is capable of simultaneously generating temporally synchronized audio and video from text. The core of SyncFlow is the proposed dual-diffusion-transformer (d-DiT) architecture, which enables joint video and audio modelling with proper information fusion. To efficiently manage the computational cost of joint audio and video modelling, SyncFlow utilizes a multi-stage training strategy that separates video and audio learning before joint fine-tuning. Our empirical evaluations demonstrate that SyncFlow produces audio and video outputs that are more correlated than baseline methods with significantly enhanced audio quality and audio-visual correspondence. Moreover, we demonstrate strong zero-shot capabilities of SyncFlow, including zero-shot video-to-audio generation and adaptation to novel video resolutions without further training.

## 1 Introduction

Humans experience the world multimodally, where audio and video are naturally related, providing complementary information that enhances perception and understanding. This natural synchronization is reflected in most media content we consume, such as movies, virtual reality content, and human-computer interfaces. With the advancement of artificial intelligence generated content (AIGC), there has been substantial progress in generating audio or video from textual descriptions. State-of-the-art models have achieved impressive results in tasks such as image generation (Rombach et al., 2022; Ramesh et al., 2021), audio generation (Yang et al., 2023; Liu et al., 2023a; 2024a; Kreuk et al., 2022), video generation (Singer et al., 2022b; Ho et al., 2022; OpenAI, 2024b), and audio-visual cross-modal generation (Iashin & Rahtu, 2021; Luo et al., 2024; Mei et al., 2023; Mo et al., 2024), showcasing the potential of AIGC in creating realistic and engaging content.

Despite the strong correlation between audio and video, most existing AIGC research treats audio and video generation as isolated tasks, generating each modality independently (Żelaszczyk & Mańdziuk, 2022; Park et al., 2022; Yariv et al., 2024). For instance, diffusion models have recently shown potential as real-time game engines by predicting frames sequentially (Valevski et al., 2024), but audio is still not incorporated into the generation process despite the crucial role of audio in enhancing the immersive and engaging experience in gaming. While there are a few studies that explore joint audio-video generation, such as MMDiffusion (Ruan et al., 2023) and the more recent AV-DiT (Wang et al., 2024), these approaches are primarily designed for unconditional generation and are often domain-specific, such as focusing on dancing video (Li et al., 2021) or natural scenes (Lee et al., 2022). Notably, MMDiffusion offers examples of open-domain, unconditional joint audio-video generation but lacks evaluation metrics or comparative results in its publication, leaving a gap in assessing its effectiveness. Whether audio and video can be generated simultaneously from text using a unified approach has received limited attention.

Two main approaches have emerged that bring us closer to joint text-to-audio-video (T2AV) generation, though each comes with its limitations. One approach involves employing two separate

systems, such as concatenating a text-to-video (T2V) model with a video-to-audio (V2A) model or equipping a video understanding model with a text-to-audio (T2A) model (Chen et al., 2024a). While these cascaded systems can generate both modalities, they introduce additional latency and the risk of error propagation during the cascaded processing. Moreover, the lack of direct interaction between the three modalities in such systems can potentially lead to sub-optimal results. Another approach leverages a contrastively aligned latent space to generate audio and video jointly. For instance, models like composable diffusion (CoDi) (Tang et al., 2024b) align visual, audio, and textual representations in a shared latent space for T2AV generation. Similarly, the model proposed by Xing et al. (2024) adopts pretrained Imagebind (Girdhar et al., 2023), a model that aligns six modalities with contrastive learning to guide the generation of audio and video. However, these methods are limited by using a one-dimensional contrastive representation, which contains limited temporal information. Some previous work (Tang et al., 2024b) even targeted audio and video with different durations, resulting in poor temporal alignment between the modalities. Recently, TVGBench (Mao et al., 2024) addresses a text-to-audible-video generation task, which marks the first attempt on the text conditioning audio-video joint generation.

This paper introduces SyncFlow, a model capable of generating temporally synchronized audio and video from text. We propose a dual-diffusion-transformer (d-DiT) architecture to handle the synchronized generation of video and audio. The d-DiT builds upon the Diffusion Transformer (DiT) (Peebles & Xie, 2023a) architecture, which has demonstrated strong performance in both video and image generation (Esser et al., 2024; OpenAI, 2024b). To address the challenges of computational cost and the scarcity of paired audio-video data, we propose a modality-decoupled multi-stage training strategy. Specifically, we decouple the model training on video and audio before joint audio-video fine tuning. Starting with a pre-trained text-to-video model, we freeze the video generation component and adapt it to audio generation by leveraging intermediate features from the video model as conditioning inputs for audio synthesis. This decoupled approach allows the video generation related parameters to be trained using widely available text-video datasets, while the audio component can be adapted with a relatively small amount of paired data. Given the high computational demands of video generation, particularly for high-resolution and high-frame-rate outputs, our strategy significantly reduces the computational overhead of joint training and mitigates the need for large-scale text-video-audio datasets. Finally, the entire d-DiT model is finetuned end-to-end on both video and audio modalities to enhance the generation quality. Both the audio and video generation components of SyncFlow are built using a flow-matching latent generative model (Lipman et al., 2022). Our experiment shows SyncFlow not only achieves temporally synchronized T2AV but also achieves strong performance compared with cascaded systems and systems built with contrastive encoders. In summary, our contributions are as follows:

- We introduce SyncFlow for synchronized joint video-audio generation from text (T2AV). SyncFlow can jointly generate temporally synchronized 16 FPS video and 48kHz sampling rate audio with open-domain text conditions.

- We empirically show that SyncFlow performs better than other T2AV systems based on cascaded processing and multi-modal contrastive encoders.

- The pretrained SyncFlow model demonstrates strong zero-shot performance on video-to-audio generation and a zero-shot adaptation ability to new video resolutions for joint audio-video generation.

## 2  RELATED WORKS

**Rectifier Flow Matching** Flow matching (FM) (Lipman et al., 2022) is a powerful method for generative modelling that enables efficient training of continuous normalizing flows (CNFs) (Papamakarios et al., 2021) by directly predicting vector fields along fixed conditional probability paths. Building on FM, rectified flow matching (RFM) (Liu et al., 2023b) enforces straight sampling trajectories between prior and target data distributions. This process also shares a similar intuition as optimal-transport-flow (Onken et al., 2021). Compared with diffusion-based methods (Ho et al., 2020), RFM demonstrates improved sample quality on image generation while reducing the number of sampling steps (Lipman et al., 2022). Subsequent works have expanded the use of RFM to various applications, such as text-to-image generation (Esser et al., 2024; Liu et al., 2024b), point cloud generation (Wu et al., 2023a), text-to-speech synthesis (Guo et al., 2024; Mehta et al., 2024), source separation (Yuan

et al., 2024) and sound generation (Vyas et al., 2023; Prajwal et al., 2024), highlighting the versatility of RFM across different domains.

**Text-conditioned Generative Modeling** Recent years have witnessed remarkable progress in text-conditioned generative modelling. For text-to-image generation, models such as DALL-E 2 (Ramesh et al., 2022) and Stable Diffusion Series (Rombach et al., 2022) demonstrated strong performance by producing high-quality images aligned with the textual inputs. In the audio domain, there has been substantial advancement in generating speech, music, and environmental sounds from text or transcriptions (Tan et al., 2022; Liu et al., 2024a; Chen et al., 2024b; Ye et al., 2024; Li et al., 2024; Agostinelli et al., 2023; Copet et al., 2023; Huang et al., 2023). In video generation, CogVideo (Hong et al., 2023) and Make-a-Video (Singer et al., 2022a) have demonstrated early success by effectively adapting text-to-image methodologies to video through language models and diffusion-based approaches, respectively. Later, the diffusion-transformer architecture (Peebles & Xie, 2023b) has further enhanced video generation capabilities, as showcased by the OpenAI release of Sora (OpenAI, 2024b). The recently proposed CogVideoX (Yang et al., 2024) scales the open-source video generation model to 5-billion parameters, achieving state-of-the-art performance. SyncFlow differs from prior work in that it focuses on the joint generation of synchronized audio and video, posing challenges in both computational efficiency and coordination between modalities.

**Joint Audio-visual Generation** While significant progress has been made in generating audio, video, and images independently, the task of simultaneously generating audio and video from text remains underexplored. Although CoDi-2 (Tang et al., 2024a) demonstrates the ability to generate video frames and sound from text instructions, it does not directly address the T2AV task. MMDiffusion (Ruan et al., 2023), AV-DiT (Wang et al., 2024), TAVDiffusion (Mao et al., 2024) and Hayakawa et al. (2024) have demonstrated success in the joint generation of videos with accompanying audio. Some approaches perform text-conditioned joint audio-video generation by relying on contrastive modality encoders, as seen in CoDi (Tang et al., 2024b) and (Xing et al., 2024), where a shared video-audio-text contrastive-aligned one-dimensional representation is used to condition both audio and video generation. While this allows for joint generation, the one-dimensional contrastive representation lacks sufficient temporal information, limiting the performance of the model on temporal synchronization. In fact, the audio and video samples generated by CoDi[1] exhibit mismatched duration, falling short of achieving true synchronization.

## 3 METHOD

**Problem Definition** This section introduces the implementation of *SyncFlow* for jointly generating video frames $y^V = \{y_1^V, y_2^V, \ldots, y_N^V\}$ and corresponding audio samples $y^A = \{y_1^A, y_2^A, \ldots, y_M^A\}$ given a text input $s$, where $N$ is the number of video frames and $M$ is the number of audio samples. Both outputs are generated simultaneously to ensure temporal alignment between the video and audio. The video frames $y^V$ are tensors of shape $\mathbb{R}^{F \times C \times H \times W}$, where $F$ is the number of frames, $C$ are the RGB channels, and $H$ and $W$ are the height and width of each frame, respectively. The audio samples $y^A$ are monophonic, represented as a vector of shape $\mathbb{R}^M$, where $M$ is the length of the audio signal in samples. The generative process is defined as $\mathcal{G}(s; \Theta) \to (y^V, y^A)$, where $\mathcal{G}(s; \Theta)$ represents the joint generative function conditioned on the text input $s$, and $\Theta$ are the model trainable parameters. Sections 3.1 and 3.2 provide a detailed explanation of the implementation of the function $\mathcal{G}$. Section 3.1 presents the high-level overview of the proposed method, introducing the concept of latent rectifier flow matching (RFM), constructing latent spaces for both video and audio and applying RFM for joint video and audio generation. In Section 3.2, we detail the design of the dual-diffusion-transformer (d-DiT) architecture, elaborating on how it processes the latent variables of both modalities. Additionally, Section 3.2 outlines the flow-matching loss function used to optimize the model for synchronized multimodal generation and the formulation of classifier-free guidance (Ho & Salimans, 2021) we used during inference.

### 3.1 LATENT RECTIFIER FLOW MATCHING

**Preliminary: Rectifier Flow Matching (RFM)** The training of SyncFlow is based on rectifier flow matching (Liu et al., 2023b), which improves upon flow matching (Lipman et al., 2022) by optimizing

---

[1]https://github.com/microsoft/i-Code/tree/main/i-Code-V3

the transport between the prior distribution $p_0$ and the target distribution $p_1$. Given a training data sample $x_1 \sim p_1$ from the target distribution and a sample from the prior distribution $x_0 \sim p_0$, the target velocity field $v$ of RFM is calculated as $v = x_1 - x_0$. This velocity field represents the optimal direction for transporting the sample $x_0$ to the sample $x_1$ along a straight trajectory. We follow Tong et al. (2024) to perform mini-batch optimal transport within the batch of $x_0$ and $x_1$ during training to find an approximate solution to the dynamic optimal transport.

To ensure that the transport follows a straight path between the prior and the target distributions, RFM enforces that each point on this trajectory predicts the same velocity field. The intermediate points along the trajectory are determined by the forward process of the RFM, where the noisy latent variable at time $t \in [0, 1]$ is given by $x_t = (1 - t)x_0 + tx_1$. At each time step $t$, given the latent sample $x_t$, the RFM model $u(x_t, t; \theta)$ is optimized toward predicting the velocity field that minimizes the deviations with the target velocity field $v = x_1 - x_0$, in which $\theta$ are the trainable parameters for RFM. With a pretrained velocity field prediction model $u(x_t, t; \theta)$, the sampling process of RFM is obtained by solving the ordinary differential equations (ODE) $\frac{dx_t}{dt} = u(x_t, t; \theta)$, where the generative sampling process can be formulated as

$$\hat{x}_1 = x_0 + \int_0^1 u(x_t, t; \theta) \, dt. \tag{1}$$

In practice, the integral in Equation (1) is discretized into $N$ sampling steps for numerical approximation. The multiple sampling steps of RFM break down the complex generative process into smaller, more manageable steps, facilitating more accurate generation compared with directly generating samples with one step, which intuitively aligns with the inference-time scaling laws (Snell et al., 2024), as recently demonstrated by the *OpenAI-o1* model (OpenAI, 2024a).

**Latent Representation for Video and Audio** Raw video and audio data often have extremely large dimensionality. This results in high computational complexity during model training and inference, particularly when dealing with high video frame rates (FPS) and audio sampling rates. To efficiently model the high-dimensional $y^V$ and $y^A$, we adopt a latent modelling approach inspired by the latent diffusion model (Rombach et al., 2022). On both video and audio modalities, we train variational autoencoders (VAE) (Kingma & Welling, 2014) with a latent space with compressed dimensions compared with the original video or audio. The latent encodings for video and audio are formulated as $z^V = \mathcal{E}_{\text{video}}(y^V) \in \mathbb{R}^{F' \times C \times H' \times W'}$, and $z^A = \mathcal{E}_{\text{audio}}(y^A) \in \mathbb{R}^{T \times D_A}$, where $\mathcal{E}_{\text{video}}$ and $\mathcal{E}_{\text{audio}}$ are pre-trained VAE encoders for video and audio, respectively. The video encoder $\mathcal{E}_{\text{video}}$, based on a video spatial-temporal VAE proposed by Zheng et al. (2024), compresses the high-dimensional frames $y^V$ into a latent representation $z^V$ with reduced dimension on both spatial and temporal axes. The audio encoder $\mathcal{E}_{\text{audio}}$ is derived from the Encodec (Défossez et al., 2023), which was originally designed for learning discrete audio latent representation. We adapt Encodec by removing vector quantization layers and adding a kullback–leibler (KL) divergence loss to regularize the variance of the latent space following the training losses used in a standard VAE models (Kingma & Welling, 2014).

Both the video VAE and the audio VAE are paired with corresponding decoders that map the latent representations back to their original high-dimensional spaces. Specifically, the video decoder $\mathcal{D}^V$ reconstructs the video frames from the latent space $z^V$, while the audio decoder $\mathcal{D}^A$ reconstructs the audio samples from the latent space $z^A$. The decoding processes can be described as $\hat{y}^V = \mathcal{D}^V(z^V)$, $\hat{y}^A = \mathcal{D}^A(z^A)$. This ensures that the compressed latent variables can be converted back to full-resolution video and audio outputs after generation in the latent space.

**Latent Rectifier Flow Matching** The objective of SyncFlow is to generate the video and audio data from a unified perspective from the text. The core idea of SyncFlow can be formulated as

$$(\hat{v}_t^A, \hat{v}_t^V) = u(z_t^V, z_t^A, t, s; \theta), \tag{2}$$

where $z_t^V$ and $z_t^A$ represent the video and audio latent variables at time $t$, and $u(\cdot)$ is the function that predicts the velocity fields for both modalities, conditioned on the noisy latents, text conditions $s$, and time $t$. Similar to the $u(z_t, t; \theta)$ used in Equation (1), the predicted velocity fields $\hat{v}_t^A$ and $\hat{v}_t^V$ can be used to sample $\hat{z}_1^A$ and $\hat{z}_1^A$ by solving the ODE, followed by decoding through the VAE decoders to obtain the final generation output. Section 3.2 introduces the implementation of $u(\cdot)$ in Equation (2).

## 3.2 DUAL-DIFFUSION TRANSFORMER

The input variables of $u(\cdot)$ in Equation (2), including the noisy video latent $z_t^V$ and noisy audio latent $z_t^A$ differ in shape, for which we design a dual-diffusion-transformer (d-DiT) architecture, as shown in Figure 1. The d-DiT comprises two distinct towers (i.e., stacks of layers) for handling video and audio data, with a modality adaptor facilitating information sharing from the video tower to the audio tower.

**Video Generation Tower** The video latent $z_t^V$ is first processed by a three-dimensional convolutional network, which expands its channel dimension to match the embedding dimension $E$. Subsequently, the convolutional outputs are spatially split into $2 \times 2$ patches, resulting in a tensor of shape $B \times (T_v \times S) \times E_v$, where $B$ is the batch size, $T_v$ is the video latent temporal dimension size, $S$ is the number of spatial patches, and $E_v$ is the embedding dimension.

To capture both per-frame spatial information and temporal dynamics, each layer in the video generation tower includes a spatial attention layer and a temporal attention layer. Although both attention mechanisms share the same architecture, they differ in how the input is reshaped before performing self-attention (Vaswani et al., 2017). For spatial attention, self-attention is applied to the patches within each frame independently, by combining the temporal and batch di-

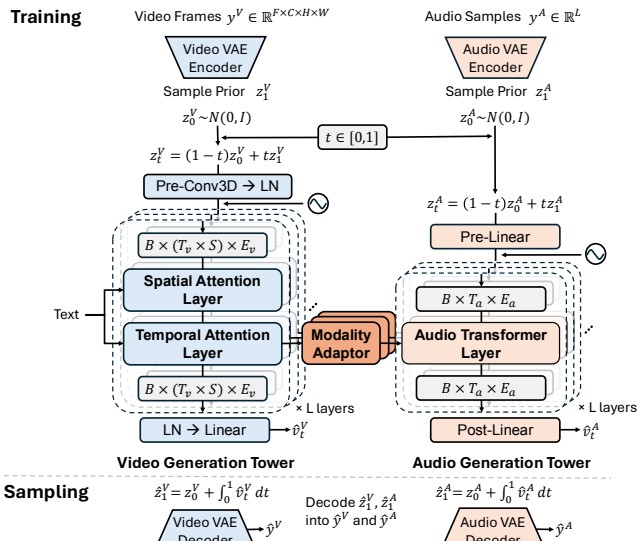

Figure 1: The main architecture of dual-diffusion-transformer (d-DiT) used by SyncFlow. Two parallel towers handle video and audio generation, with modality adaptors to enhance synchronization. Text input conditions the video generation towers through cross-attentions.

mensions of the self-attention input into a tensor of shape $(B \times T_v) \times S \times E_v$. In temporal attention, the spatial patches are combined with the batch dimension before input, and self-attention is applied to the temporal sequence, yielding a tensor of shape $(B \times S) \times T_v \times E_v$. To incorporate text-based control, we use the T5 text encoder (Raffel et al., 2020) to extract rich semantic embeddings from the input text. The encoder part of T5, pre-trained on a variety of language tasks, produces textual embeddings that are injected into both the spatial and temporal attention layers via cross-attention.

**Audio Generation Tower** The audio generation tower has the same number of layers in parallel with the video generation tower. The input of the audio generation tower has shape $B \times T_a \times E_a$, where $T_a$ and $E_a$ are the temporal dimensions and embedding dimension of the audio VAE latent, respectively. Following the architecture used in AudioBox (Vyas et al., 2023), each audio transformer layer includes 16-head self-attention, cross-attention, and a feed-forward MLP, with layer normalization applied after each transformation. The output of each temporal attention layer in the video tower is denoted as $\mathcal{F}_{\text{video}}^{(l)}$, and is used as conditioning information for the corresponding audio transformer layer. We use the output from the temporal attention layers as conditioning information, rather than the spatial attention layers, as they potentially contain richer temporal information.

**Modality Adaptor** Instead of directly using $\mathcal{F}_{\text{video}}^{(l)}$ as key and value inputs in the cross-attention operation of the audio transformer layer, $\mathcal{F}_{\text{video}}^{(l)}$ passes through a modality adaptor. This adaptor transforms the intermediate video features to ensure they are optimally suited for interacting with the audio transformer. As shown in Figure 2, the default modality adaptor we used includes a multi-head self-attention layer, followed by layer normalization and linear transformation. Our experiment indicates the adaptor helps the model to achieve lower validation loss and better metrics score (see Figure 6 and Table 4).

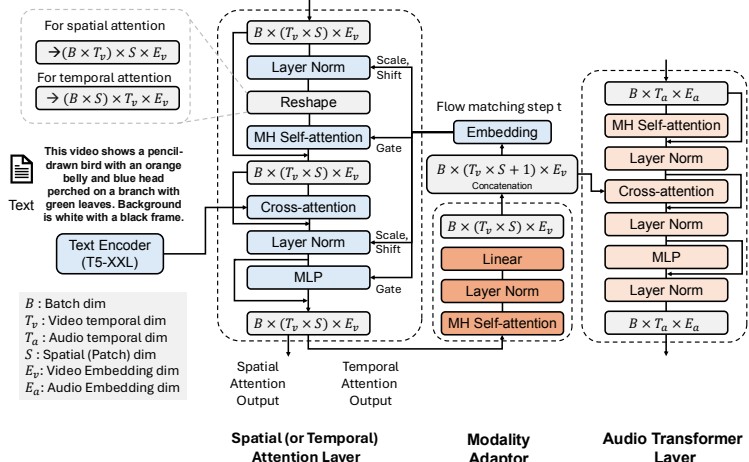

Figure 2: The detailed implementation of the spatial-temporal attention layers, audio transformer layers, and modality adaptor. The output of the modality adaptor is concatenated with the flow matching time step embedding as the cross-attention condition to the audio transformer layer.

To optimize the d-DiT architecture, the flow-matching loss is defined as the mean square error (MSE) loss between the target velocity field and model prediction, given by $\mathcal{L}_{\text{fm}} = \mathbb{E}_{t \sim \mathcal{U}(0,1)} \left[ \| u(z_t^V, z_t^A, t, s; \theta) - (v_t^V, v_t^A) \|^2 \right]$, where $v_t^V = z_1^V - z_0^V$ and $v_t^A = z_1^A - z_0^A$ represent the target velocities of the video and audio latent spaces, respectively.

During sampling, we employ classifier-free guidance (CFG) (Ho & Salimans, 2021), which has been shown to be a helpful technique to enhance the generation quality and the relevancy to the text conditions (Liu et al., 2023a). With the formulation of CFG, the final velocity prediction becomes a combination of conditional and unconditional velocity prediction.

$$(\hat{v}_t^A, \hat{v}_t^V) = \hat{u}(z_t^V, z_t^A, t; \theta) = u(z_t^V, z_t^A, t; \theta) + w \cdot \left( u(z_t^V, z_t^A, t, s; \theta) - u(z_t^V, z_t^A, t; \theta) \right), \quad (3)$$

where $w$ is the CFG guidance weight. The effects of CFG are explored in the ablation studies.

**Modality-decoupled Multi-stage Learning** Generative modelling of video and audio data is computationally intensive. To address this, we propose a modality-decoupled training strategy consisting of three stages: (1) Pretraining the video tower on text-video paired data; (2) Adapting the pretrained video tower for audio generation, where the audio tower is trained while the video tower remains frozen; (3) Jointly fine-tuning both the video and audio towers on the full training set. This approach offers two main advantages. Due to the scarcity of text-video-audio data, our method is data-efficient as it allows the video tower to be pretrained separately. Second, this method is computationally efficient. Since the video tower is frozen during the second stage, the audio tower can be trained with larger batch size, reducing computational overhead while improving performance. Experiments on audio generation can be conducted more efficiently without retraining the video tower.

## 4 EXPERIMENTAL SETUP

**Dataset** We conduct experiments using the curated VGGSound (Chen et al., 2020), and the Greatest Hits dataset (Owens et al., 2016). VGGSound was initially built using a specifically designed pipeline to ensure strong audio-video correspondence and to filter out samples with significant ambient noise. The Greatest Hits dataset (Owens et al., 2016) contains 977 videos of various objects being hit, scratched, or poked with a drumstick, capturing interactions with materials such as metal, plastic, cloth, and gravel. Each video includes both visual and audio data, making it ideal for studying the correspondence between physical interactions and their resulting sounds. We split each video in the Greatest Hits dataset into 10-second segments, resulting in a total of 2,995 segments, from which we select 744 segments as the test set, ensuring that no test samples originate from the same videos used for training. For both the VGGSound and the Greatest Hits dataset, we use the VideoOFA (Chen et al., 2023) to generate video captions automatically. We primarily use the Greatest Hits dataset for ablation studies as it is smaller in scale.

**Evaluation Metrics** To evaluate the quality of the generated video and audio, we use Fréchet video distance (FVD) and Fréchet audio distance (FAD). FVD measures the similarity between the distribution of generated and real videos by comparing feature representations extracted from a pre-trained I3D model (Carreira & Zisserman, 2017), while FAD compares generated and real audio using features from the VGGish model (Hershey et al., 2017). To assess the similarity between the generated audio and target audio, we use KL divergence (Kreuk et al., 2022), which measures the divergence between the VGGish classification output of paired audio samples. Additionally, we use the CLAP score (Wu et al., 2023b) to measure the alignment between the generated audio and the input text caption. To further examine the relationship between video, audio, and text, we employ ImageBind (Girdhar et al., 2023) to extract contrastive embeddings from each modality and calculate cosine similarity, referred to as the ImageBind Score (IB) (Mei et al., 2023). For instance, IB (Gen-A&Gen-V) denotes the IB score between the generated audio and video, which is similar to the AVHScore metrics proposed in Mao et al. (2024).

**Setup Details** We randomly sample two-second video-audio segments from the training dataset, using 16 FPS video data with centre cropping and resizing to a resolution of $256 \times 256$. The audio data are sampled at $48$kHz. The Video VAE downsamples the temporal dimension by a factor of $4$ and the spatial dimensions by a factor of $8$. We flatten the output of Pre-Conv3D (see Figure 1) into a sequence of tensors by $2 \times 2$ patch splitting on the spatial dimension. Building upon the Encodec, the audio VAE downsamples the temporal dimension by a factor of 960, resulting in an audio latent with a temporal resolution of $50$Hz and an embedding dimension of $1142$. The video generation tower utilizes a pretrained text-to-video generation model OpenSora [2]. Both video and audio generation towers in d-DiT have 28 layers and a transformer feature dimension of $1142$. The video VAE and audio VAE are pre-trained independently and remain frozen during the SyncFlow training. For the VGGSound dataset, we train the audio generation tower with a batch size of 16 per GPU for $150,000$ steps on 32 H100 GPUs, taking about $140$ hours. Joint fine-tuning of the audio and video towers is done with a batch size of 2 per GPU for $20,000$ steps. On the smaller Greatest Hits dataset, we train for $25,000$ steps with a batch size of 16 on 8 H100 GPUs. We set the CFG weight in Equation (3) to $6.0$ by default and use $50$ sampling steps during generation. Additionally, we randomly drop the text conditioning with a $10\%$ probability during training to enable CFG.

**Baselines** For the cascaded model baselines, we combine the OpenSora model with three publicly available T2A models: AudioLDM (Liu et al., 2023a), AudioLDM 2 (Liu et al., 2024a), and AudioGen (Kreuk et al., 2022), two publicly available V2A models: SpecVQGAN (Iashin & Rahtu, 2021), Diff-Foley (Luo et al., 2024), and our reproduction of another V2A model FoleyGen (Mei et al., 2023). The latter three V2A models are also compared against SyncFlow in video-to-audio generation tasks. The FoleyGen we used is our reproduced version following the original paper (Mei et al., 2023). AudioLDM is a latent diffusion model designed for generating audio from text, while AudioLDM 2 improves upon this by incorporating self-supervised pretraining for better audio quality and diversity. AudioGen frames audio generation as a conditional language modelling task, using transformer architectures to produce audio from textual descriptions. SpecVQGAN utilizes a vector-quantized autoencoder to learn compact and meaningful audio representations, combined with a transformer decoder for V2A generative modeling. Diff-Foley leverages diffusion models to create realistic sound effects for videos. As the work that perform T2AV generation using contrastively pretrained encoders, we reproduce the result of CoDi for comparison. In the original CoDi training, the video duration is 2 seconds while the audio duration is 10 seconds. For evaluation, we trim the CoDi audio generation output to the first 2 seconds to match our setup.

## 5 RESULT

Table 1 presents the evaluation results on the VGGSound dataset, including cascaded systems, CoDi, and various SyncFlow configurations trained on the VGGSound training set. In the SyncFlow-VGG setup, the pretrained video generation tower is frozen, and only the audio generation tower and modality adaptors are optimized. SyncFlow-VGG$_{128\times128}$ evaluates the pretrained SyncFlow on a different target video resolution ($128 \times 128$), which SyncFlow-VGG was not explicitly trained on. SyncFlow-VGG-AV-FT involves joint fine-tuning of both the audio and video towers, with

---

[2] https://github.com/hpcaitech/Open-Sora

Table 1: Performance evaluation of the proposed method on the VGGSound evaluation set. *Gen-V* denote the generated video. *GT-V* and *GT-T* means the ground truth video and text in the evaluation set, respectively. [†] denote the zero-shot setting.

| Setting | FAD ↓ | KL ↓ | CLAP ↑ | IB Generated Audio | | |
| --- | --- | --- | --- | --- | --- | --- |
| | | | | &Gen-V ↑ | &GT-V ↑ | &GT-T ↑ |
| GroundTruth | 0.0 | 0.0 | 0.275 | 0.276 | 0.276 | 0.200 |
| OpenSora + AudioLDM | 6.56 | 3.29 | 0.248 | 0.104 | 0.078 | 0.103 |
| OpenSora + AudioLDM 2 | 8.61 | 3.08 | 0.255 | 0.137 | 0.116 | 0.107 |
| OpenSora + AudioGen | 7.13 | 3.38 | 0.246 | 0.094 | 0.100 | 0.100 |
| OpenSora + SpecVQGAN | 5.50 | 3.06 | 0.172 | 0.086 | 0.084 | 0.065 |
| OpenSora + Diff-Foley | 10.58 | 4.30 | 0.185 | 0.144 | 0.097 | 0.085 |
| OpenSora + FoleyGen | 3.69 | 3.08 | 0.228 | 0.159 | 0.122 | 0.141 |
| CoDi | 8.44 | 3.53 | 0.174 | 0.091 | 0.084 | 0.100 |
| SyncFlow-VGG | **1.81** | **2.53** | **0.311** | 0.182 | **0.180** | **0.176** |
| [†]SyncFlow-VGG$_{128 \times 128}$ | 3.18 | 2.65 | 0.300 | 0.148 | 0.170 | 0.158 |
| [†]SyncFlow-VGG$_{512 \times 512}$ | 2.59 | 2.60 | 0.266 | 0.165 | 0.149 | 0.157 |
| SyncFlow-VGG-AV-FT | 2.36 | 2.54 | 0.308 | **0.190** | 0.178 | 0.174 |

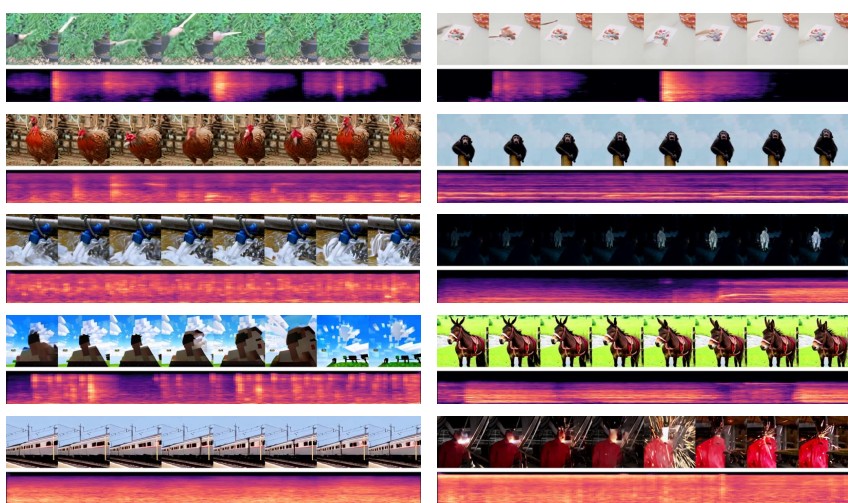

Figure 3: Snapshot of video and audio generated by SyncFlow. The video frames are displayed every four frames for simplicity. The original audio frame length corresponding to each audio is 32.

parameters initialized from SyncFlow-VGG. Based on the experimental results, we can draw the following conclusions.

*The proposed system outperforms the cascaded methods.* The cascaded methods include both OpenSora followed by V2A models and OpenSora followed by T2A models. The cascaded systems exhibit significant variation in performance, with the best-performing OpenSora+FoleyGen achieving an FAD score of 3.69. Notably, the three T2A-based cascaded systems demonstrate worse FAD scores, likely due to the lack of fine-tuning on the VGGSound dataset, which leads to a gap in the data distribution. Moreover, the audio generated by SyncFlow-VGG variants generally achieves a higher IB score with the generated video than the ground truth video, despite the latter typically having better overall video quality. This suggests that sharing information between the video and audio towers during generation helps the audio adapt to video-specific characteristics, such as acoustic environment, gender, and distance. Also, we found joint fine-tuning of the audio and video towers improves synchronization. As observed with SyncFlow-VGG-AV-FT, after jointly fine-tuning both towers with smaller batch sizes, the system exhibits better ImageBind scores between the generated audio and video, indicating improved synchronization. We show examples of SyncFlow-VGG generation in Figure 3.

*Cascaded methods are prone to error propagation.* The absence of interaction across all three modalities in cascaded systems introduces the potential for error propagation. This is evident in the lower CLAP and IB (Gen-A & Gen-V) scores. While T2A-based systems generally achieve higher CLAP scores, their IB (Gen-A & Gen-V) scores are lower than V2A-based systems, suggesting that T2A models lack sufficient conditioning from the visual modality, and V2A models lack conditioning from the text modality. This supports the hypothesis that cascaded systems are prone to error propagation, leading to suboptimal results.

*Our proposed system outperforms the modality contrastive encoder-based system.* Since CoDi conditions its audio and video generation modules on a one-dimensional vector without sufficient temporal information, it is reasonable that it delivers suboptimal performance on the T2AV task.

Besides, SyncFlow can generate videos at new target resolutions along with corresponding audio, as seen with SyncFlow-VGG$_{512 \times 512}$ and SyncFlow-VGG$_{128 \times 128}$. The IB (Gen-A & Gen-V) score for the $512$ resolution is higher than for the $128$, while the CLAP score is lower. This suggests that a higher video resolution may have more influence on cross-attention conditions than text conditions.

**Video Generation Tower Performance** Table 2 compares the video generation quality across different settings. Overall, the pretrained OpenSora performs significantly better than CoDi, which is developed based on the Make-a-Video model (Singer et al., 2022b). Besides, results show that increasing the target resolution in the pretrained OpenSora model leads to improved performance, with the $512 \times 512$ resolution achieving the best IB score. Notably, OpenSora sometimes achieves higher ImageBind scores than ground truth video-caption pairs. This discrepancy may arise from imperfections in the video captioning model, VideoOFA, which sometimes assigns captions that do not fully align with the video content. In contrast, the generated videos, being directly conditioned on these captions, can potentially achieve better alignment. After fine-tuning the video generation tower, SyncFlow-VGG-AV-FT achieves the best FVD score, indicating that fine-tuning on the training set helps align the model target space with the data distribution in the evaluation set.

Table 2: Performance comparison on different video generation pipelines.

| Model | Resolution | FPS | FVD ↓ | IB Gen-V & GT-T ↑ | IB Gen-V & GT-A ↑ |
|---|---|---|---|---|---|
| GroundTruth | $256 \times 256$ | 16 | 0.6 | 0.332 | 0.276 |
| CoDi | $256 \times 256$ | 4 | 718.8 | 0.338 | 0.181 |
| OpenSora | $128 \times 128$ | 16 | 506.9 | 0.308 | 0.164 |
| OpenSora | $256 \times 256$ | 16 | 397.7 | 0.356 | 0.203 |
| OpenSora | $512 \times 512$ | 16 | 331.8 | **0.374** | 0.209 |
| SyncFlow-VGG-AV-FT | $256 \times 256$ | 16 | **298.4** | 0.350 | **0.220** |

**Zero-shot Video-to-Audio Generation** Diffusion and flow-matching-based generative models have proven effective in tasks like in-filling and out-painting (Liu et al., 2023a; Rombach et al., 2022), where part of the target data is known. In these cases, noise is added to the known information, replacing the predicted part of the model, so that each denoising step incorporates the noisy version of the ground truth. This process, often referred to as latent inversion (Lan et al., 2024), can also be applied to editing tasks, where denoising begins with partially noisy data, and the process is guided by specific editing instructions. Similarly, SyncFlow can perform V2A generation by replacing the predicted video latent $\hat{z}_t^V$ with the ground truth latent $z_t^V$, ensuring that the model receives accurate guidance from the ground truth video at each denoising step.

Table 3 presents the video-to-audio (V2A) performance across different systems. SyncFlow demonstrates competitive results compared to other approaches. When comparing these results with Table 1, where SyncFlow-VGG achieves a KL divergence of $2.53$ and an IB (Gen-A & Gen-V) score of $0.182$, introducing ground truth video information into the generation process leads to significant improvements in both metrics. In the V2A setting, the IB (Gen-A & Gen-V) score increases to $0.210$, indicating the T2AV system potential upper bound if video generation is well-aligned with ground truth. Figure 4 shows examples of SyncFlow on the zero-shot V2A generation.

**Ablation Studies** Our ablation study addresses two key questions: (1) How important is the modality adaptor? and (2) How effectively does the audio tower utilize video information from the video generation tower? To address the first question, we conduct an experiment where features from the video generation tower are directly used as conditions for audio generation, bypassing the modality

Table 3: Performance comparison on zero-shot video-to-audio generation. The FoleyGen[†] is the internel version.

| Setting | Zero-shot | FAD ↓ | KL ↓ | IB Gen-A & GT-V ↑ |
|---------|-----------|-------|------|-------------------|
| GroundTruth | - | 0.0 | 0.0 | 0.276 |
| FoleyGen† | ✗ | 1.31 | 1.97 | 0.297 |
| SpecVQGAN | ✗ | 5.53 | 2.91 | 0.094 |
| Diff-Foley | ✗ | 7.85 | 3.51 | 0.143 |
| SyncFlow-VGG | ✓ | 1.81 | 2.44 | 0.210 |

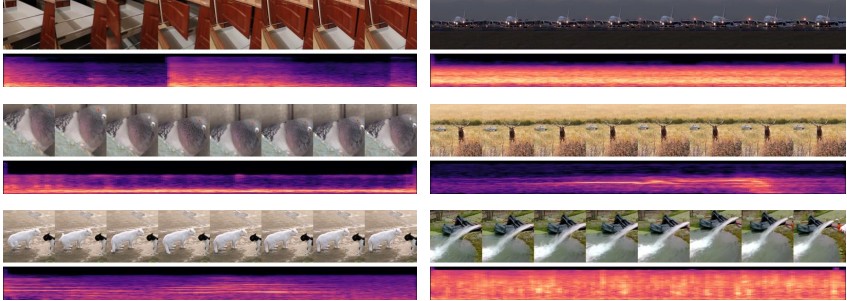

Figure 4: Example of zero-shot video-to-audio generation using SyncFlow. The input video is sourced from the VGGSound evaluation set.

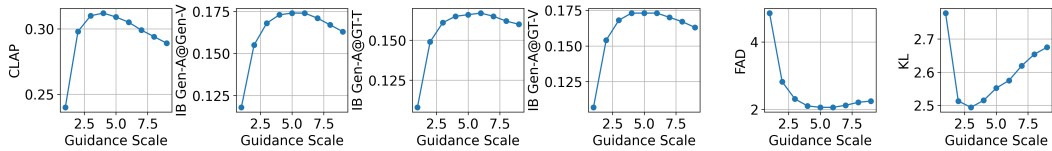

Figure 5: The effect of classifier-free guidance scale on the performance of SyncFlow-VGG.

Table 4: Ablation studies on the Greatest Hits dataset.

| Setting | FAD ↓ | IB Gen A & Gen V ↑ | CLAP ↑ | KL ↓ |
|---------|-------|--------------------|--------|------|
| SyncFlow-GH | **0.92** | **0.156** | 0.354 | 2.88 |
| SyncFlow-GH *w/o* modality adaptor | 2.34 | 0.144 | 0.313 | 3.37 |
| Text-to-audio *w/* SyncFlow Audio Tower | 1.01 | 0.138 | **0.379** | **2.74** |

adaptor. This configuration, referred to as *SyncFlow-GH w/o modality adaptor*, is designed to assess the impact of incorporating a modality adaptor before conditioning the audio generation tower. For the second question, we evaluate text-to-audio generation by using the audio tower without any video information, ensuring that the model relies solely on the text embeddings extracted by the T5 text encoder. Given the relatively small scale of the Greatest Hits dataset, we report the average performance of the last three checkpoints (saved every 500 training step) for more reliable results.

As shown in Table 4, removing the modality adaptor results in a noticeable performance drop compared to SyncFlow-GH, highlighting the importance of the adaptor in improving synchronization between audio and video. Also, Figure 6 in the Appendix A.2 shows that with and without the modality adaptor can have a clear gap in the validation loss. In the text-to-audio generation setting, the model achieves better CLAP and KL scores, but the IB score, which indicates the audio-video correspondence, shows a clear degradation. This suggests that while text-based audio generation can lead to better text-audio alignment (CLAP), incorporating video information during the generation process significantly enhances synchronization between the audio and video modalities. We also perform ablation studies on the best classifier guidance scale to use, which is shown in Figure 5. Not all metrics show the same trend with the change of the guidance scale. We chose 6.0 as the default guidance scale as it has the best average IB score.

# 6 CONCLUSIONS

In this paper, we introduced SyncFlow, a model for joint audio and video generation from text, addressing the limitations of existing cascaded and contrastive encoder-based methods. By leveraging the dual-diffusion-transformer (d-DiT) architecture and a modality-decoupled training strategy, SyncFlow efficiently generates temporally synchronized audio and video with improved quality and alignment. Our experiments demonstrated strong performance on multiple benchmarks, including VGGSound and Greatest Hits, showcasing the ability of SyncFlow to achieve strong audio-visual correspondence and zero-shot adaptability to new video resolutions. Additionally, our ablation studies highlighted the importance of the modality adaptor in enhancing synchronization between modalities.

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

# A APPENDIX

## A.1 LIMITATIONS

The model is trained on a video sub-clip randomly sampled from a video in the dataset, while the text caption from VideoOFA is based on the full-length video. This means the caption we used for the model training is not optimal. Nevertheless, most videos have consistent semantics, so our model generally works fine. Improving caption quality could be a way to improve the proposed method.

Despite carefully curating the VGGSound dataset, we still observe videos with static frames and ambient sounds, such as videos with static album covers or food-sizzling sounds. There are also a lot of off-screen sounds in the data, such as narration, environmental sound, etc. Future work can be done to address the data quality issue, such as filtering the data based on audio-visual correspondence.

The samples generated by the text-to-video model can sometimes lack clear and coherent movement, leading to potential ambiguities and mismatches during training and inference. Future work could focus on enhancing the performance of the video generation tower to address these issues.

## A.2 FIGURES

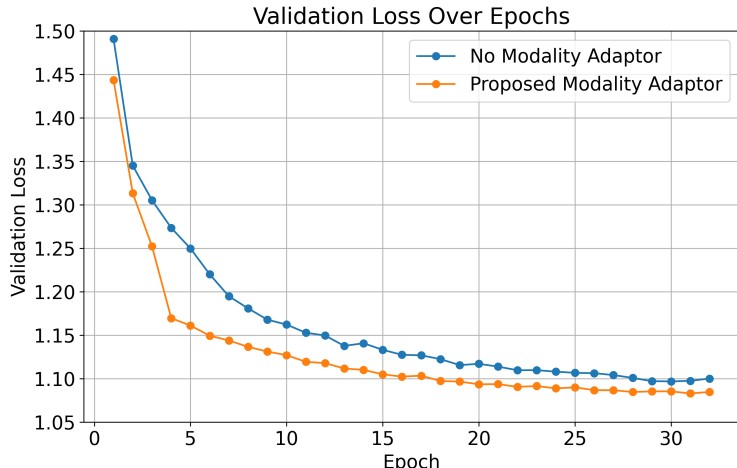

Figure 6: Validation loss on the Greatest Hit dataset with and without the proposed modality adaptor.

