# OpenReview forum: "SyncFlow: Temporally Aligned Joint Audio-Video Generation from Text"
_ICLR.cc/2025/Conference — Submitted to ICLR 2025_

### Official Review · Reviewer_y3DN · 2024-11-02

**Soundness:** 3
**Presentation:** 2
**Contribution:** 3
**Rating:** 5
**Confidence:** 4

**Summary:**

This paper presents SyncFlow for joint audio-video generation from a given text prompt.
To this goal, this paper introduces a dual diffusion transformer (d-DiT) and modality-decoupled multi-stage training strategy.
SyncFlow can jointly generate 16fps videos and 48kHz audio.
The authors show the superiority on the vggsound and greatest hits datasets, and compare the model with the cascaded system.

**Strengths:**

- The modality-decoupled multi-stage training strategy is a practical solution for training a dual diffusion transformer.
- The results show that SyncFlow outperforms the cascaded systems and codi.

**Weaknesses:**

- Why do we need the RFM for this task? The authors should explain the motivation and validity of this choice. As in the Introduction section, the authors only explain that 'Both the audio and video generation components of SyncFlow are built using a flow-matching latent generative model'.

- Why don't we need an audio-to-video adapter? Moreover, we can consider an interactional two-way adapter. There is no reason or ablation study for this.

- It's hard to understand why the authors chose anonymized models such as T2A-1, T2A-2, etc for the evaluation part. There exist open-source text-to-audio models. I think it is unreasonable to compare the performance of the joint model without knowing the configuration and basic performance of the models.

- As the authors mentioned in 524-526, text-based audio generation shows more powerful text-audio alignment, as shown in CLAP score, why SyncFlow-GH works better than SyncFlow-GH w/o modality adapter in CLAP and KL score? The results are still confusing.

- As the authors tackled, imagebind is not good for representing temporal features. So, using another av-align score [1] or synchronized model [2] to evaluate audiovisual synchronized performance.

- I understand that the author values the task of generating audio and video based on text, but for comparison of generation quality with existing simultaneous generation methods (especially to demonstrate the efficiency and effectiveness of the proposed structure, this is not a text-specific module and structure) performance comparison in landscape or aist++ is required (using null text)

[1] Diverse and Aligned Audio-to-Video Generation via Text-to-Video Model Adaptation
[2] PEAVS: Perceptual Evaluation of Audio-Visual Synchrony Grounded in Viewers’ Opinion Scores / https://github.com/amazon-science/avgen-eval-toolkit

**Questions:**

-  Are the base models used in the cascaded model finetuned with vggsound? Is it fair to compare this with the SyncFlow trained on vggsound?

---

> ### Author Response · Authors · 2024-11-27
>
> Thank you for your detailed review and constructive feedback. Below, we address your concerns and questions point by point:
>
> ### Weaknesses:
> 1. **Motivation and Validity of RFM for this Task:**
> The use of Rectified Flow Matching (RFM) is motivated by recent advancements in the flow matching model on image, video and audio generation. As RFM encourage straight trajectory, it usually result in better sample quality with less sampling steps. Another important motivation for us to use RFM is the pretrained text-to-video model we used, OpenSora-v1.2, is trained with RFM. To maintain formulation consistency we choose also to use RFM.
>
> 2. **Why No Audio-to-Video Adapter or Two-Way Adapter:**
> Our goal is to adapt the pretrained OpenSora model (via multi-stage training) to simultaneously generate video and audio while keeping modifications to the video generation tower minimal. To achieve this, we introduced a unidirectional modality adaptor for video-to-audio fusion, ensuring the video tower remains largely unchanged.
>
> 3. **Use of Anonymized Models in Evaluation:**
>    We understand the concern regarding anonymized model names in the initial submission. To address this, we have de-anonymized the baseline models in the revised manuscript as follows:
>    - **T2A-1:** AudioLDM 1
>    - **T2A-2:** AudioLDM 2
>    - **V2A-1:** SpecVQGAN
>    - **V2A-2:** DiffFoley
>    - **T2VGen:** OpenSora
>
>    These updates clarify the configurations and performance of the baseline models. Furthermore, these baselines were chosen based on their reproducibility and widespread use in text-to-audio or video-to-audio tasks.
>
> 4. **Confusion Regarding SyncFlow-GH w/ and w/o Modality Adaptor:**
> First of all please allow me to clarify the setting of "SyncFlow-GH w/o Modality Adaptor": In this configuration, features from the video generation tower are directly used as conditions for the audio generation tower, bypassing the modality adaptor. This setup evaluates the impact of the modality adaptor on the performance of audio generation and audio-visual alignment. SyncFlow-GH with the modality adaptor performs better in CLAP and KL scores because the adaptor effectively enhances the conditioning from video tower (which also includes text information) to audio tower. This design ensures that the audio generation process leverages the processed and potentially more optimal conditioning information.
>
> 5. **Temporal Alignment Metrics:**
>    We acknowledge that ImageBind, while useful, is not optimal for evaluating temporal alignment. As suggested, metrics like Onset Accuracy and AV-align scores provide more robust evaluations of synchronization. Due to time constraints during the review period, we were unable to implement these metrics but plan to incorporate them in future evaluations.
>
> 6. **Performance Comparisons on AIST++ or Landscape Datasets:**
> This is an important direction for future work, and we will acknowledge this limitation in the updated manuscript.
>
> ---
>
> ### Questions:
> 1. **Fairness of Cascaded Model Comparisons:**
>    Most of the V2A baseline models used in our experiments, such as SpecVQGAN and Diff-Foley, are trained or fine-tuned on the VGGSound dataset. While part of the T2A baseline models we employed were not specifically fine-tuned on VGGSound, they still included VGGSound as part of their training data. We apologize for any confusion caused by the anonymization of these models in the initial submission, which may have obscured these details.
>
> ---
>
> Thank you once again for your valuable feedback, which has helped us improve the clarity and quality of our work. We hope these clarifications address your concerns and provide a better understanding of SyncFlow’s contributions and design choices.

---

> ### Comment · Reviewer_y3DN · 2024-12-02
> **Thanks for the authors' response.**
>
> Thanks for the authors' response. Some concerns (anonymous models, ablation study) are solved, but I maintain the original rating.
>
> 1. Expanding an audio model with the same structure to extend the existing RFM-based video models is not considered a structural novelty. It also does not show the advantages of this structure compared to the performance of other structures.
>
> 2. I found that the main contribution of this paper is temporally synchronized video and audio generation, but the evaluation for temporal alignment is still limited. I also watched the demo site, but there was a lack of samples to evaluate this.

---

### Official Review · Reviewer_mJv8 · 2024-11-03

**Soundness:** 3
**Presentation:** 3
**Contribution:** 2
**Rating:** 5
**Confidence:** 4

**Summary:**

This paper proposed SyncFlow to simultaneously generate audio and video from text, where the dual-diffusion transformer structure and modality adapter are designed to jointly model audio and video with temporal synchronization. The empirical evaluations show that the proposed method generates better aligned video and audio.

**Strengths:**

1. The paper proposes flow-matching-based models for text-to-audio-video joint generation by designing dual diffusion transformers to handle each modality.
2. The modality-decoupled multi-stage training strategy is proposed to efficiently make use of data and improve the quality and temporal alignment between generated audio and video.
3. The paper is generally well-written.

**Weaknesses:**

1. Although the modality adapter is designed to align video with audio, there are no further explanations why just considering unidirectional (video-to-audio) fusion and how the adapter achieves the temporal alignment beyond semantic alignment.
2. It is a little confusing why text captions are only injected into the video branch rather than both video and audio branches.
3. Some experiment details are unclear and missing. For example, what are the learning rate and optimizer for (multi-stage) training? Whether the Video VAE and Audio VAE are jointly trained, individually trained, or from frozen pre-trained ones. What are the used text-to-audio (T2A-1 and T2A-1) and video-to-audio (V2A-1 and V2A-2) models?
4. It is better to present subjective performance by providing the demo or video files.

**Questions:**

1. Did you compare the proposed method with MM-Diffusion and TAVGBench discussed in the part of the related work?
2. The text caption used in the proposed method is obtained from video. It is better for T2AV to consider more suitable text captions by describing both audio and video.
3. The ImageBind score seems to only consider the semantic alignment. Did you try other metrics to evaluate the temporal alignment performance of the proposed model?
4. Frob Table 2, why the ground truth of the ImageBind score is not optimal like IB Gen-V & GT-T?

---

> ### Author Response · Authors · 2024-11-27
> **Reply to reviewer mJv8's comments**
>
> Thank you for your thoughtful review and constructive feedback. Below, we provide responses to address your comments and clarify the raised concerns:
>
> #### Weakness #1
> Our goal is to adapt the pretrained OpenSora model (via multi-stage training) to simultaneously generate video and audio while keeping modifications to the video generation tower minimal. To achieve this, we introduced a unidirectional modality adaptor for video-to-audio fusion, ensuring the video tower remains largely unchanged. The adaptor implicitly learns temporal alignment during training through the design of the model architecture, which integrates video features as conditions for audio generation. We acknowledge that better metrics for quantifying temporal alignment could further validate these improvements and consider this an important direction for future work.
>
> #### Weakness #2:
>
> The decision to use only video captions for our model is based on the following considerations:
>
> **Consistency with the Pretrained Backbone:**
>
> Our model builds on the pretrained text-to-video model OpenSora (or T2VGen in the previous anonymized version), with the goal of extending it for joint video and audio generation. OpenSora was originally trained exclusively with video captions, and maintaining this setup ensures that its video generation quality remains consistent with its pretrained performance. Adding audio captions could potentially disrupt the pretrained dynamics of the video backbone.
>
> **Encouraging Video-Dependent Audio Generation:**
>
> By excluding direct audio captions, we aim to encourage the audio generation process to rely more on visual information from the video. This approach helps avoid a scenario where the model depends heavily on audio captions for audio generation, potentially overlooking visual cues. Enhancing the reliance on video information can improve audio-visual alignment and lead to better synchronization between the two modalities.
>
>
> #### Weakness #3:
> We have updated the paper to include missing experimental details, such as:
> - Learning rate and optimizer: We use the AdamW optimizer with an initial learning rate of \(1 \times 10^{-4}\) for all training stages.
> - Training approach: The Video VAE and Audio VAE are pre-trained independently and remain frozen during the SyncFlow training stages.
> - Baseline models: The de-anonymized baseline models are as follows:
>  - **T2A-1:** AudioLDM 1
>   - **T2A-2:** AudioLDM 2
>   - **V2A-1:** SpecVQGAN
>   - **V2A-2:** DiffFoley
>   - **T2VGen:** OpenSora
>
> #### Weakness #4
> Please find our demo available here https://syncflow-core.github.io/syncflow-demo/
>
> ---
>
> #### Responses to Specific Questions:
>
> 1. **Comparison with MM-Diffusion and TAVGBench:**
>    While MM-Diffusion and TAVGBench are discussed in the related work, we did not directly compare them due to the lack of publicly available code or pretrained models for replicating their results. As MM-Diffusion does not open-source their model checkpoint pretrained on AudioSet, it is challenging to compare with MM-Diffusion on open-domain scenarios.
>
> 2. **Use of video-derived text captions:**
>    Please refer to weakness #2
>
> 3. **Temporal alignment evaluation beyond ImageBind:**
>    While we acknowledge that ImageBind is not an optimal metric for evaluating temporal synchronization, it effectively highlights clear differences between approaches. This suggests that there is still room for improvement, even in semantic alignment.
>
> 4. **Ground truth ImageBind scores in Table 2:**
>    The suboptimal ImageBind score for the ground truth data likely arises from limitations in the pre-trained video captioning model used for evaluation. Captions may not fully represent the actual content, leading to a lower semantic alignment score. We have clarified this in the revised text.

---

> > ### Comment · Reviewer_mJv8 · 2024-12-02
> >
> > Thanks for your response. I will maintain current rate due to the following reasons:
> > 1. The temporal alignment performance is not evaluated because there are some existing methods to do it, like the AV-Align [1] score.
> > 2. The proposed method is not compared with MM-Diffusion whose implementation scripts and pre-trained weights are public.
> > 3. From the given demo, the quality of the generated video and audio is not good.
> > 4. The proposed model just adopts the video caption without considering the audio caption. Although the responses explain that the usage of the video-only caption is used to be consistent with its pre-trained performance, the audio caption may provide supplementary information for benefiting both video and audio generation.
> >
> > [1] Yariv, Guy, et al. "Diverse and aligned audio-to-video generation via text-to-video model adaptation."

---

### Official Review · Reviewer_yTpv · 2024-11-03

**Soundness:** 3
**Presentation:** 3
**Contribution:** 3
**Rating:** 5
**Confidence:** 5

**Summary:**

The paper titled "SyncFlow: Temporally Aligned Joint Audio-Video Generation from Text", is capable of generating temporally synchronized audio and video from textual descriptions. The core innovation of SyncFlow lies in its dual-diffusion-transformer (d-DiT) architecture, which enables joint modeling of video and audio with proper information fusion. The system employs a multi-stage training strategy to manage computational costs, initially separating video and audio learning before joint fine-tuning. SyncFlow demonstrates enhanced audio quality and audio-visual correspondence over baseline methods and exhibits strong zero-shot capabilities, including video-to-audio generation and adaptation to new video resolutions without further training.

**Strengths:**

1. SyncFlow addresses a significant gap in the field by providing a unified approach to generate synchronized audio and video from text.
2. The dual-diffusion-transformer architecture is a creative solution that leverages the strengths of diffusion models and transformers for multimodal generation, leading to improved synchronization and quality.
3. The model's ability to perform zero-shot generation and adaptation to new video resolutions is a significant advantage, showcasing its flexibility and generalizability.
4. The designed Modality Adaptor is a reasonable strategy to connect the two modalities, and its performance is demonstrated through ablation experiments.

**Weaknesses:**

1. The paper does not fully mention existing work [1] in the process of storyline design. Existing work has systematically explored the text to audio-video generation task and designed corresponding baselines, but this paper does not mention it in the introduction chapter.
[1] Mao Y, Shen X, Zhang J, et al. TAVGBench: Benchmarking Text to Audible-Video Generation. ACM MM, 2024.
2. The dataset used in this paper is relatively small (VGGSound), and only uses VideoOFA to caption the video. No caption is seen for the audio. The model trained in this way maybe cannot properly represent the information of the two modalities.
3. The designed ImageBind-based score has been proposed in [1], and the author needs to elaborate on the difference between the two.

**Questions:**

See [Weaknesses].

---

> ### Author Response · Authors · 2024-11-26
> **Reply to reviewer yTpv's comments**
>
> We appreciate your detailed feedback and constructive suggestions. Below, we address the identified weaknesses:
>
> #### Weakness #1:
> We have revised the introduction section to incorporate the suggested reference [1].
> #### Weakness #2:
> We use only the video captions as input to the model to prevent it from overly relying on audio caption information during content generation. This design choice encourages the model to infer audio content by leveraging video information.
> #### Weakness #3:
> We have employed evaluation metrics similar to those used in the TAVGBench [1]. We have updated the evaluation section to include a citation to the recommended paper, ensuring our methodology is more aligned with established practices.
>
> [1] Mao, Yuxin, et al. "TAVGBench: Benchmarking text to audible-video generation." Proceedings of the 32nd ACM International Conference on Multimedia. 2024.

---

> > ### Comment · Reviewer_yTpv · 2024-11-26
> > **Regarding the doubts about the lack of audio caption.**
> >
> > I don’t understand why we only use video captions, why we can’t let the model rely on audio caption information. As a simultaneous generation task, audio caption as well as video caption should be used as captions.

---

> > > ### Author Response · Authors · 2024-11-26
> > > **Reply to reviewer yTpv's comments "doubts about the lack of audio caption"**
> > >
> > > Thank you for your insightful question. The decision to use only video captions for our model is based on the following considerations:
> > >
> > > 1. **Consistency with the Pretrained Backbone:**
> > >    Our model builds on the pretrained text-to-video model [OpenSora](https://github.com/hpcaitech/Open-Sora) (or T2VGen in the previous anonymized version), with the goal of extending it for joint video and audio generation. OpenSora was originally trained exclusively with video captions, and maintaining this setup ensures that its video generation quality remains consistent with its pretrained performance. Adding audio captions could potentially disrupt the pretrained dynamics of the video backbone.
> > >
> > > 2. **Encouraging Video-Dependent Audio Generation:**
> > >    By excluding direct audio captions, we aim to encourage the audio generation process to rely more on visual information from the video. This approach helps avoid a scenario where the model depends heavily on audio captions for audio generation, potentially overlooking visual cues. Enhancing the reliance on video information can improve audio-visual alignment and lead to better synchronization between the two modalities.
> > >
> > > We hope this explanation clarifies our design choices. We also recognize that using both audio and video captions as input could represent an alternative approach to designing the model. We will explore and compare these two strategies through ablation studies in our future work. Thank you again for raising this important point.

---

> > > > ### Comment · Reviewer_yTpv · 2024-11-27
> > > > **Importance of audio captions**
> > > >
> > > > Thanks for your reply. I still think that audio caption is indispensable because this task is a simultaneous generation task.
> > > > According to the author's idea, if the OpenSora weight is reused, two stages can be designed. The first stage uses video caption, and the second stage uses both audio and video caption.
> > > > In short, the author's reply did not answer my question. In any case, using only video caption does not meet the task setting.

---

### Official Review · Reviewer_4wTG · 2024-11-03

**Soundness:** 2
**Presentation:** 3
**Contribution:** 2
**Rating:** 5
**Confidence:** 4

**Summary:**

This paper introduces SyncFlow, a text-to-video-and-audio generation model that uses rectified flow within a dual Diffusion Transformer (d-DiT) architecture to synchronize audio and video outputs. To enhance efficiency, SyncFlow employs multi-stage training. Experimental results show that SyncFlow achieves competitive performance compared to baseline methods.

**Strengths:**

1. The proposed approach of using a dual-diffusion transformer for joint video-audio generation sounds reasonable.
2. Modality adaptor improves audio quality and audio-video synchronization, as verified by the lower validation loss.

**Weaknesses:**

The principal weaknesses of this work are as follows:

1. The paper lacks a sufficient comparison with relevant baselines. Existing works (T2AV / V2A), such as Seeing&Hearing (Xing et al., 2024), Diff-Foley (Luo et al., 2024), and V2A-Mapper [1], which provide replicable results on the VGGSound dataset, are not quantitatively or qualitatively compared. Including these baselines would offer a fair evaluation of SyncFlow’s effectiveness.

2. The experiments do not fully support the paper’s claims. Since the authors argue that a one-dimensional contrastive representation lacks sufficient temporal information, relying on the IB score alone seems insufficient to validate improved synchronization between audio and video. To better substantiate claims of enhanced audio-visual synchronization, a metric like onset accuracy (Owens et al., 2016) should be included in the GreatestHits (Owens et al., 2016) benchmark. Furthermore, to support the claim that multi-stage learning is efficient and beneficial, an ablation study on the computational cost of full joint training, or on the number of datasets required for additional joint training, would be required.

3. The paper lacks important details about the experimental settings. The anonymized baselines (e.g., T2VGen, T2A-1, T2A-2, V2A-1, and V2A-2) make it difficult to assess the paper's contributions in context. For instance, there is no clarity on the training data or architecture used for T2VGen + T2A-1/2, making it hard to discern if SyncFlow’s apparent superiority is simply due to additional training on VGG data, as seen in Table 1. Moreover, the details of contrastive loss are not explained, nor are the ablation study settings. For example, in Table 4, it is unclear how the text input is incorporated in settings like "SyncFlow-GH w/o modality adaptor" or "Text-to-Audio w/ SyncFlow Audio Tower."

4. A comparison in video format alongside audio would be valuable for verifying SyncFlow’s audio-video synchronization qualitatively.

[1] Wang, H., Ma, J., Pascual, S., Cartwright, R., & Cai, W. V2a-mapper: A lightweight solution for vision-to-audio generation by connecting foundation models. In AAAI 2024.

**Questions:**

1. In Table 1, why is there a performance drop in FAD, KL, and CLAP scores for SyncFlow-VGG-AV-FT compared to SyncFlow-VGG?

2. Why isn’t text information directly integrated into the audio tower? As shown in Table 4, this could improve the CLAP score. Wouldn’t incorporating text into the audio tower enhance alignment and contextual accuracy? Additionally, because audio generation depends entirely on visual input, SyncFlow seems limited in generating background sound, which could be a drawback. What are authors' thoughts on this?

3. In Table 2, how does IB Gen-V achieve a higher score than the ground truth? Is this metric reliable for evaluating alignment?

4. What exactly is the training set for the model? It appears that the VGGSound training set is used for Tables 1 and 2, while the Greatest Hits training set is used for Table 3. Why were only VGGSound and Greatest Hits chosen for training? Are these datasets considered sufficient for open-domain? Did the authors consider using larger datasets like AudioSet?

5. Unlike the authors' claim in Line 48, MM-Diffusion also conducted experiments using the open-domain dataset AudioSet. The authors need to do clearer research on related work.

6. In Figure 2, Modality Adaptor, what does the concatenation $B \times (T_v \times S + 1) \times E $ represent? Which features are being concatenated?

---

> ### Author Response · Authors · 2024-11-26
> **Reply to reviewer 4wTG's comments**
>
> Thank you for your valuable feedback and insights on our paper. We acknowledge that certain parts of our presentation may have caused misunderstandings, and we aim to address your concerns below:
>
> ## Weakness #3: Experimental settings clarity
> - Previously anonymised baseline models have been de-anonymized in the revised paper:
>   - **T2A-1:** AudioLDM 1
>   - **T2A-2:** AudioLDM 2
>   - **V2A-1:** SpecVQGAN
>   - **V2A-2:** DiffFoley
>   - **T2VGen:** OpenSora
> - The mention of contrastive loss was a typo in the text. We have clarified in the revised version that the final proposed system does not employ contrastive loss.
> - Regarding "SyncFlow-GH w/o modality adaptor", this configuration directly uses features from the video generation tower as conditions for audio generation, bypassing the modality adaptor. This setup is designed to evaluate the significance of incorporating a modality adaptor before conditioning the audio generation tower.
> - "Text-to-Audio w/ SyncFlow Audio Tower" conditions the audio generation solely on text, omitting information from the video tower, to analyze the importance of video input in enhancing audio generation.
>
> ---
>
> ## Weakness #1: Lack of comparison with baselines
> - We have compared our method with Diff-Foley (referred to as V2A-2 in the previous anonymized version) and SpecVQGAN (V2A-1). Results in Table 1 demonstrate that OpenSora+Diff-Foley does not perform comparably to our proposed model. The inclusion of SpecVQGAN further illustrates our approach's effectiveness.
> - Regarding the *Seeing & Hearing* work, the T2AV implementation code is unavailable on their GitHub repository (https://github.com/yzxing87/Seeing-and-Hearing), which limits our ability to replicate their results for comparison.
> - Similarly, for the V2A-Mapper baseline, the GitHub repository (<https://github.com/heng-hw/V2A-Mapper>) lacks inference code to reproduce their results. Therefore, direct comparisons in a way similar to how we employ Diff-Foley are infeasible in this case.
>
>
> ## Weakness #2 and #4 are addressed in later comments due to space limitation
> ---
>
> ## Responses to Specific Questions:
> Question #1 **Performance drop in Table 1 for SyncFlow-VGG-AV-FT:**
>    The drop is likely due to overfitting the video tower during fine-tuning on VGGSound. Since the video tower (OpenSora) was pre-trained on large-scale video datasets, fine-tuning it on the smaller VGGSound dataset may restrict its generated data distribution. This degradation in video quality can negatively impact audio generation, as the audio tower is conditioned on video tower features.
>
> Question #2 **Text information in the audio tower (Table 4):**
>    Incorporating text directly into the audio tower risks over-relying on text conditions while neglecting visual cues. Although this improves CLAP and KL scores, as shown in Table 4, it reduces semantic alignment (IB score) between video and audio, highlighting the importance of leveraging visual information.
>
> Question #3 **IB Gen-V exceeding ground truth (Table 2):**
>    This may stem from imperfections in the video captioning model, which lowers IB scores for ground truth videos and text. Conversely, generated videos, conditioned directly on captions, align better with the captions. Another contributing factor could be resolution effects, where higher-resolution videos can potentially yield higher IB scores.
>
> Question #4 **Choice of training datasets:**
> As highlighted in the VGGSound dataset paper (Chen et al., 2020): "VGGSound ensures audio-visual correspondence and is collected under unconstrained conditions." With over 310 sound classes, VGGSound effectively approximates open-domain settings. We opted not to perform experiments on AudioSet due to concerns about the quality of audio-video alignment in its samples. Compared to AudioSet, VGGSound is more carefully curated for audio-visual correspondence while also focusing on open-domain scenarios, making it potentially more suitable for our task. Additionally, to evaluate performance in a narrower domain, we included the Greatest Hits dataset in our experiments.
> Chen, Honglie, et al. "Vggsound: A large-scale audio-visual dataset." ICASSP 2020-2020 IEEE International Conference on Acoustics, Speech and Signal Processing (ICASSP). IEEE, 2020.
>
> Question #5 **Clarification on MM-Diffusion experiments:**
>    MM-Diffusion provides open-domain audio-video generation samples in their appendix but does not report evaluation results for comparison. We have updated our paper to clarify this.
>
> Question #6 **Clarification on Figure 2 (Modality Adaptor):**
>    The concatenation involves the modality adaptor output and flow matching step embedding \( t \). The revised caption clarifies this aspect. We have added further clarification in the caption of Figure 2.
>
> ---
>
> We hope this addresses your concerns and provides the necessary clarifications. Thank you again for your detailed review and constructive feedback.

---

> ### Comment · Reviewer_4wTG · 2024-11-26
>
> Thanks to the authors for the responses, which clarified many of my concerns.
>
> I have a follow-up question regarding the ablation studies:
>
> * **SyncFlow-GH w/o modality adaptor**: Does this setting involve passing a null input to the cross-attention layer in the Audio Tower?
> * **Text-to-audio w/ SyncFlow Audio Tower**: In this setting, does the cross-attention layer in the Audio Tower take the T5 embeddings as input?
> * For these two settings, is the model entirely retrained, with changes limited to the cross-attention layer's input shape?
>
> Additionally, I have some remaining concerns and suggestions:
>
> 1. While it is noted that Seeing & Hearing and V2A-Mapper do not provide code, I believe Seeing & Hearing can still be incorporated as a V2A baseline. Given that Seeing&Hearing employs keyframe extraction and captioning, it could address the limitation of VideoOFA relying on full-length videos, as mentioned in A.1 Limitations.
> 2. The paper provides an insufficient explanation of the baselines. A brief description of these baselines, either in the related work or methodology sections, would be helpful. Additionally, differences in the experimental setup for baselines should be clarified. For example, SpecVQGAN and Diff-Foley produce audio of different lengths, which necessitates explaining how this was managed.
> 3. As mentioned by authors in response to Question #1, fine-tuning on VGGSound tends to overfit the Video Tower due to the dataset's limited size. Therefore, the choice of training datasets (response to Question #5) should have included larger datasets. I recommend that the authors address this limitation.
>
> My weakness #2, which I believe is a critical issue, remains unaddressed. I will check the other reviews and follow up soon.

---

> ### Author Response · Authors · 2024-11-27
> **Reply to reviewer 4wTG's following-up questions**
>
> Thank you for your follow-up questions and for pointing out the areas that require further clarification. Below, we address your specific concerns regarding the ablation study settings (**The PDF update may be delayed, but we will ensure all changes are reflected in the paper before the Nov 28 deadline**):
>
> - **"SyncFlow-GH w/o Modality Adaptor":**
>   In this configuration, features from the video generation tower are directly used as conditions for the audio generation tower, bypassing the modality adaptor. This setup evaluates the impact of the modality adaptor on the performance of audio generation and audio-visual alignment. We apologize for any confusion in our earlier response and have added further clarification in the ablation study section of the revised manuscript.
>
> - **"Text-to-Audio w/ SyncFlow Audio Tower":**
>   In this setting, the cross-attention layer in the Audio Tower uses the T5 text embeddings (previously utilized as input to the video generation tower) as its condition. This evaluates the effect of using text-only conditioning on audio generation.
>
> Both configurations are retrained from scratch, with the video tower initialized from OpenSora's pretrained weights. The key differences are:
> 1. Whether the modality adaptor processes video features before conditioning the Audio Tower.
> 2. Whether audio generation uses video features or relies solely on text embeddings."
>
> ---
>
> For your remaining concerns and suggestions:
> 1. **Incorporating Seeing & Hearing as a Baseline:**
> We acknowledge that Seeing & Hearing could be a useful baseline for V2A generation. However, recent work (Zhang et al., 2024) indicates that Seeing & Hearing performs worse on V2A tasks compared to Diff-Foley, which influenced our decision not to include it as a baseline. More importantly, our experiments indicate that the fundamental limitation lies in the two-stage generation approach itself, rather than the choice of a specific V2A model. For example, even the best-performing open-source systems in a two-stage pipeline achieve an FAD of around 5.5, whereas our end-to-end system achieves a significantly lower FAD of 1.81. While incorporating the V2A component of Seeing & Hearing might provide additional insights, our current results sufficiently highlight the inherent drawbacks of two-stage systems. Nonetheless, we will consider including Seeing & Hearing as a baseline in future studies focusing on V2A models."
>
> Zhang, Yiming, et al. "Foleycrafter: Bring silent videos to life with lifelike and synchronized sounds." arXiv preprint arXiv:2407.01494 (2024).
>
> 2. **Improved Baseline Descriptions:**
> Thank you for your suggestion. We have revised the paper to provide additional explanations of the baselines, ensuring better context and understanding for readers.
>
> 3. **Choice of Training Datasets:**
>    We agree that training on larger datasets, such as AudioSet (as previously suggested), could be beneficial. However, we also emphasize the importance of data quality. While AudioSet is larger in scale compared to VGGSound, the latter is more carefully curated for audio-visual correspondence. As noted in the VGGSound dataset paper (Chen et al., 2020): *"VGGSound ensures audio-visual correspondence and is collected under unconstrained conditions."*
>
> ---
>
> We hope these answers address your concerns. Weaknesses #2 and #4 are addressed in the next part.

---

> > ### Comment · Reviewer_4wTG · 2024-11-27
> >
> > Thank you for the replies and clarifications. However, supporting the authors' claim in the response that VGGSound is more carefully curated for audio-visual correspondence is difficult. As noted in Section A.1 of the paper, the quality of the VGGSound dataset is already suboptimal. Both VGGSound and AudioSet are collected from YouTube under unconstrained conditions, and AudioSet also states [here](https://research.google.com/audioset/) that it verifies the presence of sounds within YouTube segments. If overfitting issues were observed in the video generation tower using VGGSound (as mentioned in the response to Question #1), this raises important questions about the choice of dataset and its impact on the model's performance. However, I understand that the authors may not have had sufficient time to retrain the model with a larger dataset during the discussion period.

---

> ### Author Response · Authors · 2024-11-27
> **Reply to reviewer 4wTG's following-up questions**
>
> (continued ...)
>
> # Weakness #2: Better evaluation metrics and why multi-stage learning
>
> 1. **ImageBind as a Metric for Temporal Information:**
>    While the ImageBind feature is one-dimensional and therefore not ideal for capturing temporal information, it does not entirely lack such information. For example, the AVHScore proposed by Mao et al. 2024 also use Imagebind to assess audio-video alignment, which is the evaluation method we followed. While the evaluation may not achieve perfect accuracy, it can still highlight relative trends between models and demonstrate the improvements made by our method. This makes it a useful, although not perfect, metric for evaluating audio-visual synchronization.
>
> 2. **Efficiency and Benefits of Multi-Stage Training:**
>    The efficiency of multi-stage training in our approach is evident for the following reasons:
>    - **Computational Efficiency:** Training a single text-to-video generation model, such as OpenSora-v1.2, requires over 30 million video samples and approximately 35,000 GPU hours on H100 GPUs (as per the [Open-Sora report](https://github.com/hpcaitech/Open-Sora/blob/main/docs/report_03.md)). Attempting to train both the video and audio towers from scratch without leveraging multi-stage learning would require significantly more compute resources and time, making it impractical and inefficient.
>    - **Better Utilization of Task-Specific Data:** Multi-stage training enables effective use of task-specific datasets. Text-to-video tasks typically benefit from a larger amount of high-quality data compared to audio-video data. By first training the video generation tower and subsequently training the audio generation tower, we ensure that the model takes full advantage of available data for each task, enhancing both efficiency and performance.
>
> # Weakness #4
> We have created a demo website showcasing the results of our model: [SyncFlow Demo Website](https://syncflow-core.github.io/syncflow-demo/).
>
> We hope our responses address your feedback.

---

> > ### Comment · Reviewer_4wTG · 2024-11-27
> >
> > I will adjust my score from 3 to 5 given the following unresolved issues:
> >
> > The effectiveness of SyncFlow remains difficult to evaluate. Although the authors mentioned plans to build a webpage, the paper lacks sufficient qualitative examples. Most existing video or audio generation papers include many examples to demonstrate the quality of their models, but this paper falls short, making it hard to assess the quality of the generated results.
> >
> > Regarding Weakness #2, the authors did not provide additional experimental results during the discussion period. While I acknowledge that the ImageBind score offers a measure of audio-video alignment, it is unclear why the authors did not also employ other better evaluation metrics for temporal alignment, such as Onset Accuracy/AP or the AV-align score [1], as suggested by reviewer y3DN. These metrics are widely used and simply obtainable.
> >
> > [1] Diverse and Aligned Audio-to-Video Generation via Text-to-Video Model Adaptation

---

> > > ### Author Response · Authors · 2024-11-27
> > > **Reply to reviewer 4wTG's following-up comments**
> > >
> > > Thank you for your thoughtful follow-up comments and for adjusting your score. We greatly appreciate your recognition of our efforts and your constructive feedback.
> > >
> > > **Qualitative Examples and Demo Website:**
> > >    To address concerns about the lack of qualitative examples, we have created a demo website showcasing the results of our model: [SyncFlow Demo Website](https://syncflow-core.github.io/syncflow-demo/).
> > >
> > > **On Temporal Alignment Metrics:**
> > >    We acknowledge that additional temporal alignment metrics, such as Onset Accuracy or the AV-align score (as suggested), would provide further validation of audio-video synchronization. Due to time constraints during the discussion period, we were unable to include results for these metrics. However, we recognize their importance and plan to integrate them into our evaluation framework in future work. For now, we believe the ImageBind score, while not perfect, highlights relative trends and improvements across models and provides a consistent baseline for comparison.
> > >
> > > We sincerely appreciate your constructive feedback, which has helped us improve our work. Thank you again for your thoughtful review and for giving us the opportunity to address these points.

---

### Meta-Review · Area_Chair_ALSC · 2024-12-08

**Metareview:**

This paper proposes a SyncFlow, a text-to-video-and-audio generation model. The dual Diffusion Transformer (d-DiT) and modality adapter are chosen for the design choice.

After the rebuttal period, all the reviewers reached a negative consensus. The main concerns that have not been addressed are as follows: (1) Lack of justification for the choice of RFM video models, (2) Lack of the evaluation for temporal alignment, which is the main contribution of this paper, (3) Lack of the justification of the proposed scenario, (4) Lack of the comparisons of the existing methods, and (5) insufficient qualitative results

After carefully reading the authors' responses and the reviewers' opinions, I recommend rejection for this paper, especially considering (2) and (4).

**Additional Comments On Reviewer Discussion:**

- Lack of experiments
    - Temporal Alignment Metrics: The authors couldn't provide results for Onset Accuracy and AV-align scores (which are already open-sourced). Reviewer y3DN, mJv8, and 4wTG think that reporting this metric is mandatory for this work.
    - Qualitative examples: The authors provided a demo page. However, Reviewer mJv8 argues that the quality of the generated video and audio is insufficient.
    - Comparison methods: The authors clarified that Seeing & Hearing and V2A-Mapper were not publically available, but they reported Diff-Foley and SpecVQGAN. However, MM-Diffusion has public implementation and pre-trained weights (according to Reviewer mJv8).
- Motivation of RFM
    - According to the authors' rebuttal, the main reason for RFM is that the chosen pre-trained T2V model, OpenSora-v1.2, uses RFM. As Reviewer y3DN mentioned, the advantage of RFM against the other possible candidates remains unclear.
- Motivation for the choice of adapter design
    - The authors clarify that the goal of this paper is to adapt the pre-trained OpenSora-v1.2 model with minimal changes.
- No audio caption is used but only video caption is used
    - The authors mentioned that it is because OpenSora is chosen as a base architecture, and OpenSora was originally used with video captions without audio captions. They also mentioned that they intended to encourage video-dependent audio generation. However, Reviewer mJv8 and yTpv didn't agree with this argument, but they still argued audio caption would be beneficial.

There were more discussions between the authors and the reviewers, not mentioned here. However, they were minor compared to the above discussions or simple clarification.

---

### Decision · Program_Chairs · 2025-01-22

Reject